# Position: We Fool Ourselves Thinking the 'X' in XAI is Useful

## Abstract

This position paper argues that the prevailing XAI paradigm suffers not from technical limitations but from a profound philosophical misconception: the assumption that explanation is primarily about transparency. We argue that the development of useful explanation is not fundamentally a question of information transfer, but of epistemic parity. Through a case study of harbor monitoring, we demonstrate that although stakeholders are without doubt conceptually misaligned—misalignment primarily results from clashes between domain-specific assumptions and authority claims. We call for replacing transparency-focused XAI with a new paradigm of Domain Authority Negotiation (DAN) that explicitly acknowledges explanation as a contestation of epistemic power rather than a mechanism of revelation.

## 1 Introduction

We fool ourselves thinking that the X in XAI is useful—at least, not for making AI systems acceptable to all relevant stakeholders. In reality, "X" more often reflects the priorities of system designers than the epistemic needs of those affected. Despite a proliferation of techniques, metrics, and toolkits promising "explainability," most current AI research offers explanations that satisfy neither practical accountability nor meaningful understanding. Heatmaps, saliency scores, and attribution graphs provide a veneer of interpretability—legible to developers, but largely irrelevant, and often misleading, to end-users, legal experts, and ethical deliberation. **This paper argues that the prevailing technical paradigms in XAI fail not only in execution but in their fundamental conception: they mischaracterize explanation as a formalizable information problem and overlook the critical issue of epistemic authority**.

At the heart of this failure lies the profound misconception that with sufficient mathematical refinement, technical explanations can satisfy all stakeholders. In reality, usefulness requires establishing an equilibrium between multiple perspectives, purposes, and demands. This in turn requires realizing an ideal of 'epistemic parity'—a state in which all relevant perspectives are taken duly into account. When stakeholders from different domains request explanations, they enter a hierarchical struggle over whose perspectives matter most. So far, technical disciplines have won the battle for AI validation authority—but the rest of society has lost.

Interdisciplinary collaboration is inherently challenging, partly because it involves navigating not just different vocabularies but competing authority frameworks. Researchers are trained—and disciplined—to assert their domain's authority over specific aspects of knowledge production. Below conceptual misalignment lie (sometimes subconscious) issues of professional identity and mutual disrespect, creating persistent conceptual misalignment that undermines effective collaboration [1; 2; 3; 4].

Submitted to 39th Conference on Neural Information Processing Systems (NeurIPS 2025). Do not distribute.

Although 'explanation' may be defined and used in superficially similar ways across disciplines, its function varies dramatically based on the authority it serves. Technical explanations assert the primacy of mathematical validation; legal explanations assert the primacy of doctrinal interpretation; organizational explanations assert the primacy of governance alignment; and ethical explanations assert the primacy of value frameworks. These positions cannot be reconciled through improved technical methods alone because they represent fundamentally incompatible authority claims.

This authority struggle manifests in policy efforts like the EU guidelines for trustworthy AI [5] and the recently enacted AI Act [6], which mandate "transparency" requirements. Yet these mandates often obscure the underlying authority contests. By framing explanation as a technical transparency problem rather than an issue of balanced epistemic authority, they perpetuate the illusion that refined mathematics can satisfy all stakeholders simultaneously.

The basic XAI challenge demands not only technical solutions, but explanations that can be interpreted meaningfully across all relevant contexts of use. However, the nuances of these interdisciplinary verification challenges can only be fully understood through careful examination of concrete applications.

In this paper, we present an approach to revealing multi-domain misalignment on AI validation. Based on a case study, we propose Domain Authority Negotiation (DAN) as a radical alternative to current XAI paradigms. Rather than chasing the unicorn of universal mathematical explainability, DAN explicitly acknowledges explanation as an authority negotiation problem and provides mechanisms for adjudicating competing institutional claims. Only by confronting the epistemic power dynamics at the heart of explanation can we move beyond the current impasse end in XAI research.

## 1.1 The alternative view

Influential XAI research overwhelmingly treats "explanation" as a mathematical object: defined through axioms, formal properties, and algorithmic criteria that can be computed, proved, or optimized [e.g. 7; 8; 9; 10; 11; 12]. These works ground explanations in game theory, logic, and function approximation—largely severed from, e.g., social context, legal reasoning, or ethical stakes. The focus is on formal tractability, soundness, and axiomatic neatness, with stakeholder perspectives at best an afterthought.

Relying solely on axiomatic formalization as the foundation for explanation is naïve. Such system-centered views overlook what truly makes explanations enlightening and actionable in practice. This raises a critical question: can the mathematically sanitized notion of "explanation" capture the complex, normative roles explanations serve in science and society?

To further challenge this view, we briefly consider the philosophical question: **what is an explanation?** Simply put, explanations answer why something happens. There are at least two fundamental types of explanatory answers to 'why' questions [13; 14]. First, causal explanations trace the mechanisms that bring about a phenomenon—physical processes, computational operations, or environmental conditions producing an outcome. Second, intentional explanations account for behavior by detailing reasons, goals, or intentions—what agents aim to achieve given their beliefs and desires.

Both types may be relevant in XAI but are often conflated. Moreover, each type can be articulated at different levels. For example, a causal explanation of a model's output might range from high-level system's purpose, through specific procedures or models, down to physical or architectural details [15]. Similarly, intentional explanations in human-agent interaction can vary from detailed reasoning to general purpose summaries.

The nature of causality itself is complex and debated. Naïve monocausal models suggest single causes deterministically produce effects. While intuitive, this approach fails to capture real-world intricacies. More nuanced frameworks view causality as involving enabling and triggering conditions, where effects depend on groups of individually insufficient but non-redundant conditions, forming part of a collectively unnecessary but sufficient set of causes [16]. Contemporary discussions further emphasize causality's probabilistic nature, where causes increase outcome likelihood without guaranteeing it [17].

In XAI, what counts as a "cause" varies by context and theory, shaping explanations, fairness assessments, and trust. Crucially, an explanation's appropriateness depends on its intended purpose and must align with contextual needs [18]. Explanations serve diverse goals—from engineering

debugging, to regulatory compliance, to encouraging trust in medical advice. Given AI's varied deployment contexts, this challenge is fundamental yet underappreciated.

## 2 Case: Machine Unlearning in Harbor Front Monitoring

In this paper, then, we gauge the depth of conceptual misalignment through a systematic, qualitative study of how experts from different fields interpret XAI outputs. We compare different disciplinary conceptualizations of a concrete, published example of explainability and examine the extent to which visualizations and metrics are considered explanatory from different perspectives. What appears initially as mere "conceptual misalignment" we interpret as a deeper contest over epistemic authority—each field implicitly positioning its own standards as the benchmark for what counts as a valid explanation.

For this purpose, we focus on a recently published paper [19] that employs XAI techniques to explore "machine unlearning". The work focused on a use case in harbor front monitoring, where thermal cameras track and count various objects. The specific challenge was to explain an AI systems ability to forget how to categorize humans while maintaining its ability to identify other objects like vehicles and bicycles.

In this real-world application, the paper leveraged SIDU (Similarity Difference and Uniqueness) [20], an attribution-based XAI method, to verify the effectiveness of machine unlearning. SIDU generates heatmaps, visual representations that show which parts of an image the AI system pays attention to when making decisions, enabling the tracking of focus shifts after unlearning. From a technical perspective, these heatmaps revealed fascinating patterns: after unlearning, the system's attention was visibly shifted away from human patterns while maintaining focus on other objects (see Fig. 1 in appendices).

To provide rigorous verification of the unlearning process, quantitative measures were developed. By introducing two complementary metrics—Heatmap Coverage (HC) and Attention Shift (AS)—the model quantitatively tracks the success of unlearning. Together, these metrics provide a quantitative means of verifying successful unlearning, complementing the qualitative insights from heatmap visualizations. Fig. 2 in the appendice illustrates this process by showing how HC and AS metrics can be used to evaluate the effectiveness of different unlearning methods.

The SIDU visualization method operates by computing attribution scores for each pixel in the input image, highlighting regions that significantly influence the model's decision. The HC metric quantifies spatial attention alignment by calculating the intersection between attention heatmaps and ground-truth object locations, while the AS metric measures the Euclidean distance between attention distributions before and after unlearning. In technical evaluations, successful unlearning was indicated by HC values decreasing significantly for human regions (average reduction of 62.3%) while remaining stable for non-human objects (average change of ±7.5%), and AS values exceeding a threshold of 0.5 for images containing humans. From a computer vision perspective, these quantitative metrics provide mathematical rigor and meaningfully explain how the algorithm shifts its attention between different types of objects.

During discussions of this work in an interdisciplinary setting, however, a notable observation was made: while technical experts found the visualizations and metrics persuasive, experts in organizational studies, philosophy, and law deemed them far from convincing as tools of explanation. This aligns with previous research on misalignment between design intentions and actual stakeholder needs [e.g., 21; 22].

### 2.1 Methodology

To study this phenomenon, we developed a qualitative mixed methods approach that combines case study analysis, expert elicitation, and comparative interpretation. While our approach is primarily qualitative, it provides empirical grounding through structured expert assessment of a real-world XAI application rather than theoretical conjecture. Consequently, this practice-based investigation captures active, real-world interpretative processes across disciplines.

Our methodological design deliberately places different disciplinary perspectives on equal footing—an explicit attempt to counter the epistemic hierarchies that typically privilege technical vali-

dation over other domains. This methodological choice itself represents a challenge to the implicit authority claims that shape interdisciplinary XAI research.

This approach follows a structured three-phase process:

1. **Case Selection and Analysis**: The case was selected as a concrete instance where XAI methods are applied for explanatory purposes.

2. **Expert Elicitation**: Domain experts were invited to evaluate the case example from their disciplinary perspective. Contributors were sampled based on established expertise in their respective fields and previous experience with the AI domain. Identical questions were posed to each expert to ensure comparability.

3. **Cross-disciplinary Comparative Analysis**: Responses were analyzed using a qualitative framework that examined (a) the conceptual role of explanations in each discipline, (b) criteria for evaluating XAI methods, and (c) identified gaps between technical capabilities and disciplinary requirements.

Specifically, we invited experts to address the following research questions based on their close reading of the original XAI paper:

**RQ1** What role do explanations play in your field?

**RQ2** To what extent do the presented visualizations and metrics meet the criteria relevant to your discipline?

**RQ3** What, if anything, is missing from these methods to make them valuable?

These questions were posed to senior experts in organizational studies, law, and ethics, all of whom are involved in ongoing interdisciplinary projects related to responsible AI development. Each expert was asked to write a two-page response from their own disciplinary perspective. This resulted in three highly distinct responses.

In our analysis, it became clear that their distinct writing styles and academic orientations were not incidental, but integral to their differences. Rather than merely reflecting separate domains of expertise, these stylistic and methodological choices actively shape the perspectives they advance. The very form of their responses—analytical frameworks, evidentiary standards, and rhetorical strategies—embody their respective domain's authority claims. Legal experts orient themselves towards legal doctrine as the ultimate arbiter; ethicists position ethical frameworks as the proper validation standard; and organizational scholars center governance structures as the relevant authority.

Consequently, we decided to reproduce the contributions in their original form to give the reader the full experience of their differing approaches (see appendices). Moreover, our analysis revealed that these disciplinary variations reflect implicit epistemic authority claims—each domain asserting its own framework as the rightful arbiter of what constitutes adequate explanation.

## 3 Results

This section presents our findings on how scholars from organizational studies, law, and ethics interpret XAI explanations in the context of the harbor-front monitoring case. We begin with brief interpretive commentaries on each disciplinary response, highlighting key features of their respective perspectives. These insights are then synthesized in a comparative analysis that connects back to our broader research questions.

Rather than viewing these responses simply as parallel viewpoints, we analyze them as manifestations of competing authority claims—each discipline implicitly asserting its right to determine what constitutes valid explanation and validation in AI. Seen through this lens of epistemic power dynamics, the evident conceptual misalignment appears less like a technical disagreement and more like an unresolved struggle over whose standards should prevail

It's important to acknowledge that two pages offer very limited space to engage with such complex issues. Despite this constraint, the responses provide valuable insights, as discussed below. Moreover, the concise format has proven to be both practical and actionable—offering a viable template for future iterations of the method.

## 3.1 The governance perspective

First, the response from the governance perspective provides a compelling foundation for discussing explainability in XAI cases. The text notes that in scenarios such as harbor-front monitoring, explainability must serve a highly diverse group of stakeholders—including rescue teams, municipalities, researchers, industry, and citizens. Governance concerns, thus, extend far beyond algorithmic transparency to encompass policy alignment, decision authority, trust-building, and other factors.

Against this background, the text raises a hypothetical risk that XAI explanations may fall short of meeting all governance requirements but it does not explore how different stakeholders may have distinct needs for explanation.

Particularly interesting is the text's emphasis on explainability as a means of trust-building—something fundamentally different from explainability as a quantitative measure of attention shifts in AI models. However, this crucial disconnect is only implied rather than explicitly discussed. Furthermore, the text proposes a dynamic, process-oriented view of explainability—capturing hopes, fears, setbacks, uncertainties, and opportunities—which contrasts sharply with the static, technical approach commonly found in XAI research.

As such, while the heatmap study aimed to be useful for practitioners, the organizational studies response suggests that many key governance dimensions were overlooked. From this perspective, if the goal was practical relevance, the original study should have engaged more directly with real-world governance concerns.

Although the text does not explicitly address misalignment between XAI explanations and governance needs, it strongly implies that the relevance of heatmaps and machine unlearning in governance is limited at best. The text concludes with a series of broad questions for future research, but these are formulated so generally that they offer little concrete guidance for technical development.

Several aspects of the organizational studies contribution—e.g., the emphasis of stakeholder diversity, the orientation towards dynamical processes, the focus on trust and governance alignment—directly challenges the technical domain's assumption of epistemic primacy, positioning organizational acceptance as a prerequisite for deployment and more important than technical validation alone.

## 3.2 The legal perspective

Second, the response from the legal perspective is a superbly written and highly engaging text. Yet, although law as a discipline is closely tethered to ethical and procedural issues, the perspectives raised by the legal expert points to entirely different issues than those flagged in the ethical, organizational, and technical contributions. For example, when discussing the harbor-front case, the text focuses on tort law—specifically individual risk behaviors or the absence of adequate safeguards. In such contexts, 'unlearning' the ability to recognize humans appears not only irrelevant, but potentially counterproductive.

The text offers a clear analysis of the role of explanation in legal settings, revealing a fundamental misalignment between legal and technical notions of explanation. While XAI often seeks to enhance transparency through tools like heatmaps, legal requirements for explanation span at least three distinct levels: factual evidence, doctrinal interpretation, and empirical analysis. This discrepancy suggests that current XAI methods may fall short of meeting legal standards, particularly under frameworks like the GDPR and the AI Act, which demand much more comprehensive justifications.

Although the text acknowledges the potential use of visualizations as evidence in legal proceedings, it does not directly engage with the harbor-front monitoring case or its associated privacy concerns. This omission highlights a broader gap between technical XAI solutions and the needs of legal practitioners operating in real-world contexts. That said, the response might have looked quite different had it come from a specialist in, for example, privacy law. Moreover, the text implies that effective legal XAI tools must go beyond visualizing model attention, to enable interpretation within established legal frameworks. This involves linking technical findings to legal principles and precedents—something current approaches have yet to achieve.

This implied requirement for legal interpretation represents a direct challenge to technical epistemic authority. The legal expert is not suggesting that technical explanations should be "improved" to

better serve legal needs, but that legal interpretive frameworks should determine explanation adequacy—a fundamental authority claim that technical domains have largely ignored. Further, by invoking regulatory frameworks like GDPR and the AI Act, the legal expert leverages institutional authority to challenge technical validation—positioning legal compliance as non-negotiable and therefore taking precedence over technical criteria.

## 3.3 The ethical perspective

Third, the response from the ethical perspective is, once again, highly engaging and thought-provoking, but also clearly differs in academic style, focus, and basic assumptions from the other contributions. The text centrally employs one of the most established tools in analytical philosophy: the thought experiment. By inviting the reader to imagine a series of hypothetical scenarios, it systematically unpacks the epistemic interests that may underlie a seemingly simple request for an explanation. Crucially, it shows how highly technical explanation methods, such as heatmaps, fail to connect with broader ethical concerns, perhaps in part because their quantitative nature is ill-suited to the kind of imaginative engagement that ethical inquiry requires.

In our reading, a key contribution of the text is its distinction between ethics-related and non-ethics-related epistemic interests in explanations—a distinction that fundamentally challenges the relevance of many current XAI approaches for most practical concerns. Through the harbor front case, the text illustrates how explanations serve different purposes depending on the epistemic interests at stake. While technical methods may satisfy a general curiosity about model behavior, they fall short of addressing ethical concerns related to privacy, surveillance, discrimination, and power dynamics.

The text defines ethics-related epistemic interests as involving "information that is constitutive of or instrumental in protecting an individual against suffering harm or unfairness, or for protecting an individual's autonomy, privacy, dignity, or some other relevant right (e.g., the right to an explanation), or for protecting other societal values such as transparency, trust, democracy, etc." Against this backdrop, heatmaps appear to have very limited value in meeting such epistemic interests. Even when they successfully indicate that human-related features were not used in classification—thereby suggesting some degree of privacy preservation—they offer little insight into the broader ethical stakes of AI decision-making.

The discussion suggests that meaningful ethical explanations should empower individuals to act upon the information they receive—to protect themselves from harm or injustice. By this standard, current XAI techniques fall far short. Obviously, this does not imply that they are entirely without value, but it underscores the need for XAI developers to be cautious in overestimating the relevance of technical explanations for addressing non-technical concerns.

The emphasis on empowerment constitutes an implicit authority claim—one that holds explanations should be judged by their capacity to enable human agency, rather than by their mathematical or statistical properties. By foregrounding empowerment, the ethicist elevates ethical evaluation above technical validation. Moreover, the use of philosophical tools—especially the thought experiment—reinforces this stance methodologically: it implicitly asserts that conceptual analysis, not computational precision, is the appropriate means for assessing explanation adequacy. In privileging human reasoning over algorithmic behavior, the ethicist challenges the technical domain's claim to set the standard for what counts as a meaningful explanation.

## 4 Discussion

After addressing how the three perspectives view XAI explanations, we note that while there is general agreement on the importance of explanations (RQ1), significant disparities emerge in how each discipline evaluate XAI tools (RQ2). Notably, all disciplines identify critical missing elements (RQ3) that must be addressed for XAI to be valuable in their respective domains. This comparison highlights the need for far more integrated approaches that can bridge the technical-practical gap while satisfying diverse stakeholder requirements.

Rather than treating the symptoms of conceptual misalignment with ever more refined technical solutions, we should address the underlying ailment directly: the unresolved and often unrecognized contest over epistemic authority. Doing so requires institutional mechanisms that explicitly acknowledge, negotiate, and balance competing claims to authority.

Although all three perspectives agree that explainability must go beyond mere technical transparency, they diverge on what should take its place. For organizations, the priority is aligning AI systems with governance objectives; for law, it is compliance and interpretive clarity; for ethics, it is empowering individuals to protect their rights and values. The analysis thus reveals not only a gap between technical outputs and stakeholder needs, but also fundamental divergences among the perspectives themselves over what should replace the technical paradigm as primary. In our view, these gaps are not simply matters of informational mismatch, but manifestations of unresolved authority struggles: All domains implicitly reject other domains' authority to determine what constitutes proper explanation.

A recurring theme is the lack of meaningful collaboration between disciplines. Organizational research calls for input from computer science to develop practical tools; legal scholarship seeks greater integration of empirical methodologies; and ethical inquiry demands a more holistic approach to XAI—one that bridges technical and societal dimensions. These interdisciplinary gaps are central to the challenge of making XAI genuinely useful. Yet bridging them requires more than improved technical methods or closer collaboration. It calls for explicit mechanisms to negotiate competing claims to epistemic authority, acknowledging that no single domain holds inherent primacy over the others.

While the different disciplines agree that current XAI approaches fall short, they also concur that the path forward lies in interdisciplinary collaboration rather than in any one field monopolizing the development of operational XAI. This consensus on collaboration masks a deeper question that remains unaddressed, however: who decides what counts as valid explanation? Without explicitly acknowledging and negotiating competing authority claims, calls for collaboration often reproduce existing epistemic hierarchies that privilege technical validation over other domains. Such collaboration will inevitably confront authority questions: Whose standards determine whether an explanation is "good enough"? How are conflicts between technical accuracy and organizational utility resolved? Who has final validation authority when domains disagree?

While our analysis centers on the harbor front monitoring case, the cross-disciplinary conceptual and authority misalignments we identify likely extend to other domains where XAI intersects with regulatory and ethical concerns, such as healthcare, finance, and autonomous transportation.

### 4.1 The Domain Authority Negotiation Framework

In response, we propose Domain Authority Negotiation (DAN) as a radical alternative framework that explicitly acknowledges and addresses the epistemic power dynamics inherent in AI explanation. Rather than asking "How can we make this model transparent?" DAN explicitly addresses three questions that current XAI frameworks systematically avoid:

1. Who should have validation authority over different aspects of AI systems?

2. How should conflicts between authority domains be resolved?

3. What institutional mechanisms are needed to negotiate authority boundaries?

DAN is founded on four elements that fundamentally challenge current XAI assumptions:

1. **Authority Domain Mapping:** Identifies which domains hold legitimate authority over different aspects of an AI system. In the harbor monitoring case, for example, technical experts have authority over computational efficiency and model performance; legal experts over GDPR compliance and evidence standards; organizational stakeholders over alignment with operational procedures; and ethicists over privacy protection and the ethical use of data.

2. **Explicit Authority Boundaries:** Clearly defines the scope and limits of authority across domains. For instance, legal experts may have primary authority over personal data handling under GDPR, with technical and ethical experts serving in consultative roles. Conversely, technical experts may lead decisions on model architecture, while legal and ethical domains retain notification or advisory input.

3. **Negotiation Mechanisms:** Establishes formal processes for managing authority claims across domains. These include:

- *Cross-Domain Translation Protocols*: Structured procedures for articulating concerns in domain-relevant language.
- *Authority Dispute Resolution*: Mechanisms for adjudicating conflicting authority claims.
- *Boundary Objects*: Shared artifacts for cross-domain intepretation, enabling coordination without consensus.

4. **Formalized Institutional Structures:** Sustained authority negotiation depends on formal institutional support. The framework calls for:
   - *Authority Councils*: Groups with representatives from each domain to oversee authority allocation and disputes.
   - *Negotiation Protocols*: Clearly documented procedures for raising, discussing, and resolving authority-related conflicts.
   - *Dynamic Adjustment Mechanisms*: Processes for revising authority boundaries in response to system evolution or contextual change.

DAN is not just an incremental improvement but a fundamental reconceptualization of explanation itself—from information transfer to authority negotiation. Only by explicitly addressing the epistemic power dynamics at the heart of explanation can we move beyond the current dead end in XAI research. Rather than presenting heatmaps to legal experts and wondering why they find them inadequate, DAN recognizes that heatmaps serve technical authority functions. It proposes creating distinct artifacts for legal authority functions, with explicitly negotiated connections between them—rather than assuming automatic translatability.

In practical terms, this means developing models that no longer aim to produce universal explanations. Instead, they would be engineered to recognize domain boundaries and invoke the appropriate validation framework for each component of a decision. For instance, when processing legally sensitive features, a model wouldn't generate heatmaps in the hope that they meet legal standards; it would activate a specialized legal validation module that applies doctrinal interpretation frameworks directly. This marks a shift from monolithic explanations to modular validation systems, where different parts of a decision are explained through domain-specific protocols—with explicit handoffs between domains when authority boundaries are crossed. Far from diminishing the role of technical XAI, this approach demands more sophisticated implementations that embed domain knowledge and authority structures directly into the design of explanation systems.

Applying DAN to our harbor monitoring case would transform how the system manages privacy concerns. Rather than relying solely on heatmaps to verify GDPR compliance in machine unlearning, the system would adopt a multifaceted validation approach. Technical evaluation would employ metrics like HC and AS—designed to assess the model's behavior within its proper domain—alongside metrics supporting other stakeholder requirements. Legal validation would use doctrinal frameworks to automatically verify privacy compliance, while organizational stakeholders would assess operational integration through their governance mechanisms. Ethical validation, then, would address privacy rights using specialized frameworks.

When the harbor monitoring system processes images containing humans, it would automatically trigger legal and ethical validation protocols, rather than attempting to translate technical validations for these domains. This approach would prevent the misalignments we observed, where technical explanations failed to satisfy stakeholders operating within different authority domains. The positive impact on surveillance governance would be substantial: clearer accountability, better protection of privacy rights, and more effective integration into public safety operations.

DAN has broad implications beyond the harbor monitoring case. In healthcare, for example, technical explanations of diagnostic algorithms face comparable difficulties in meeting clinical, legal, and ethical standards. The DAN framework could be effectively adapted by mapping authority domains across medical expertise, patient rights, regulatory compliance, and institutional governance. While this could positively transform patient care by clarifying who has authority over different aspects of AI-based diagnoses, it could, however, potentially concentrate power within established medical hierarchies if not carefully implemented.

Similarly, in financial services, criminal justice, and autonomous transportation, DAN could either democratize AI governance by making authority explicit and negotiable, or it could entrench existing power imbalances by formalizing them. The critical difference lies in how authority negotiations

are structured—whether they create genuine epistemic parity or merely codify existing hierarchies. Future research should empirically test the framework across these and other domains to develop tailored variations that respond to specific regulatory contexts while preserving the core commitment to explicit authority negotiation.

Negotiations must begin with the recognition that authority is distributed. Technical validation, legal compliance, organizational governance, and ethical assessment are each legitimate and necessary forms of validation that must be balanced rather than hierarchically ordered. The artifacts of explanation—whether technical visualizations, legal documentation, organizational procedures, or ethical reasoning—must be designed not just for clarity within domains, but for explicit coordination across them.

Implementing DAN requires more than acknowledging the challenge of authority negotiation—it demands the development of institutional structures to support it. This includes creating sustained forums where technical experts, legal scholars, organizational leaders, ethicists, and others can engage in genuine dialogue—not merely to "translate" across domains, but to negotiate the boundaries of their respective authority claims in concrete applications.

# 5 Conclusion

This position paper argues that entrenched epistemic hierarchies are at the root of the conceptual misalignments between technical approaches to explanation and the demands of legal, organizational, and ethical domains. Bridging this gap requires more than technical tweaks—it demands a radical shift in how we define, distribute, and negotiate explanatory authority. If explanations are to serve real-world needs, they must move beyond technical transparency to reflect contextual relevance, human interpretability, and normative alignment.

Technical authority claims manifest in practices that privilege quantitative metrics over qualitative assessments, demand mathematical formulations that obscure rather than illuminate, and treat technical transparency as the ultimate standard of trustworthiness. Meanwhile, other domains make competing claims. The result is a contested explanation landscape, where what counts as a "good" explanation is less about clarity or utility, and more about whose standards of authority prevail.

While the benefits of sound, interdisciplinary integration should be obvious, powerful structural and epistemic forces continue to hold it back. The above analysis exposes issues of conceptual misalignment which could, in theory, be addressed through sustained efforts at mutual understanding. But doing so would require a willingness to negotiate—and potentially compromise—one's position in the epistemic hierarchy. Only through, deliberate authority design can we hope to create AI systems that are genuinely accountable to the full range of human concerns they implicate.

Looking forward, key opportunities that DAN opens for the XAI field include:

1. Domain-specific explanation modules that recognize authority boundaries.
2. Formal boundary objects for cross-domain interpretation without assuming direct translation.
3. Institutional structures supporting ongoing authority negotiation in AI deployment.
4. Evaluation metrics for explanation effectiveness, accounting for authority differences.
5. Curricula for AI practitioners focused on epistemic authority dynamics, not just technical explanations.
6. Engage policy frameworks, such as the EU AI Act, with a focus on authority distribution rather than generic transparency.

These opportunities represent not just incremental improvements to current XAI approaches but a fundamental reconceptualization of what explanation means in AI contexts. By addressing them, we can develop truly effective explanations that serve diverse stakeholders without falling into the trap of assuming universal translatability.

Let's not fool ourselves anymore!

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

## 6 Appendices

These appendices provide supplementary materials that support our analysis of authority contestation in explanation. We first present the key visualizations that formed the basis of our technical approach to machine unlearning verification in the harbor monitoring case and were evaluated by experts, followed by their complete responses. These materials document how different disciplines interpret the same explanations through their distinct authority frameworks.

### 6.1 Figures

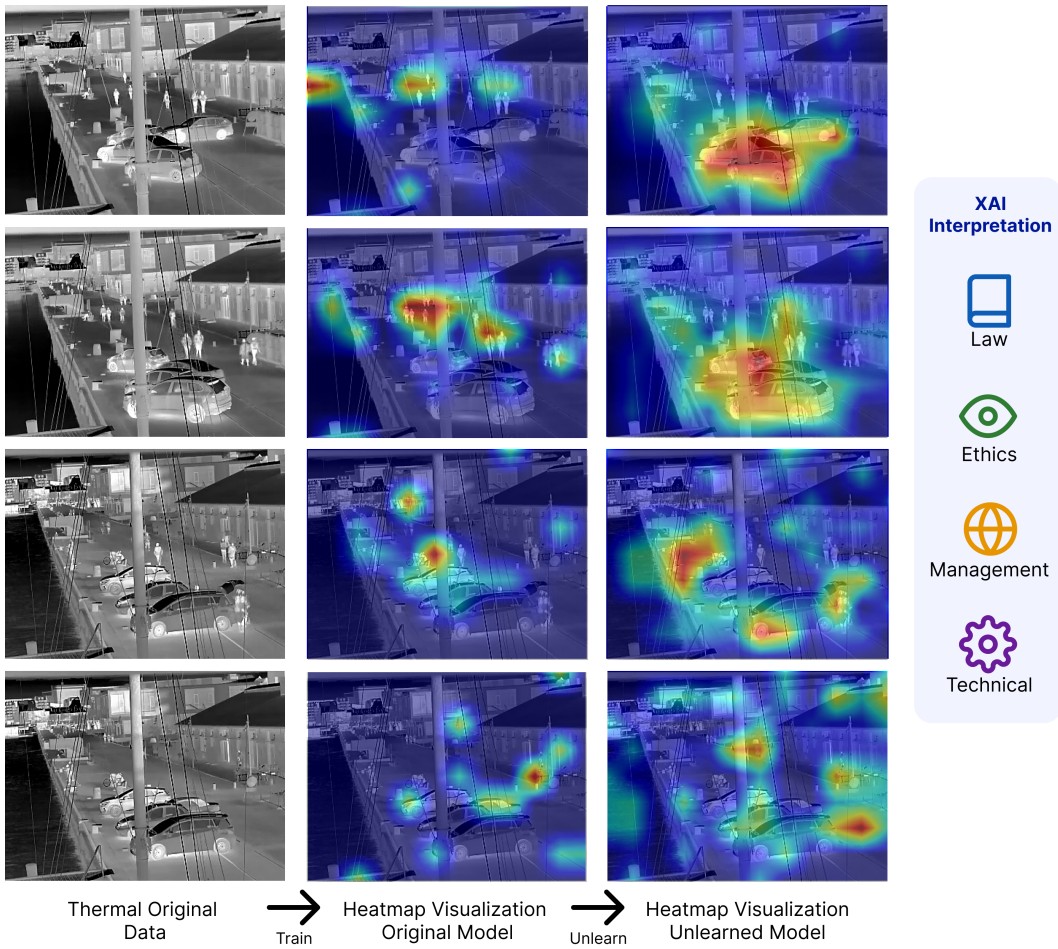

Figure 1: Evolution of model attention patterns in privacy-preserving AI: from raw thermal harbor surveillance data (left), through GDPR-violating human detection (center, red regions indicating high attention), to unlearning of human patterns while preserving other detection capabilities (right). The first two rows illustrate a successful removal of attention of human patterns, while the last two rows indicate that the unlearned model is not properly forgetting the human related regions.

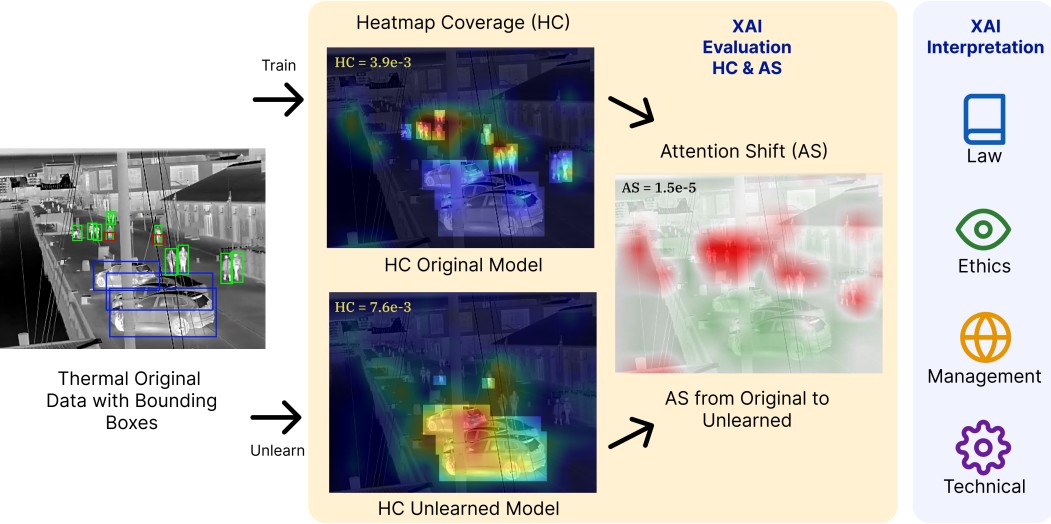

Figure 2: Our proposed verification framework utilizes XAI-based metrics (Heatmap Coverage-HC and Attention Shift-AS) to trace unlearning effectiveness. The figure shows: (left) thermal data with bounding boxes, (center) HC measurement on original model, and (right) attention shift visualization between original and unlearned models (shown via SIDU [20] difference heatmaps, where red indicates removed attention and green indicates focused attention).

## 6.2 An organizational perspective

The following section is the full text received from the invited organizational studies expert.

*While the term "explainability" originates from technological disciplines, its interpretation and application have increasingly gained traction in organizational studies due to the "black box" concern about many AI systems with sensitive personal data [23]. Explainability in technical research is primarily concerned with making AI systems' decision-making processes understandable to humans. Organizational studies adopt a broader lens on explainability that integrates technical transparency with concerns about governance, accountability, and ethical implications. Unlike the technical focus on algorithm optimization, organizational scholars view explainability as a mechanism for building trust within and outside organizational settings [24]. Organizational decision-makers must ensure that AI systems are not only technically explainable but also transparent in how decisions align with policies, public values, and legal standards. Hence, explainability in organizational contexts extends beyond technical experts to include multiple interests of stakeholders such as managers, regulators, employees, and end-users.*

*Explainability can be viewed as a bridge between AI's technical and organizational/governance dimensions. As AI becomes increasingly central to decision-making, understanding and addressing this duality will be critical for organizations aiming to use AI responsibly and effectively. However, technical metrics of machine unlearning, heatmap visualizations, and the specific metrics HC & AS are not central to the discourse or research focus in organizational studies. From this perspective, their relevance depends, among other things, on whether stakeholders can understand the logic behind heatmap visualizations and the metrics and whether they align with organizational goals or values. Although the focus on developing AI systems that can forget or ignore specific information to support privacy protection and regulatory compliance appears relevant it is also "creating the risk that the explanations provided by XAI are not the kind required for governance" [25, p. 183].*

*In the concrete use case – AI and video technology for harbor monitoring – the relevance of the proposed unlearning verification and heatmap visualizations eventually depends on how they are perceived by key actors involved in the initiative including the rescue team and the police who on an everyday basis work with harbor safety and incidents, the municipality (who sponsored the initiative); researchers (who developed the algorithm for harbor safety), hardware and software providers, and citizens. However, when studying this case from an organizational perspective, technical issues about machine unlearning and heatmaps were outside the research focus [26]. The*

*researchers aimed to understand how AI-based public safety initiatives move from ideas in research labs to real-world operations through networks of heterogeneous actors. The study showed that although a shared objective of saving lives united the involved actors, the initiative was translated from ideas to an operational AI system through a slow-paced transformation where actors dealt with hopes, fears, setbacks, uncertainties, and renewed opportunities to move the initiative forward. While technical entities such as cameras, AI prototypes, and sensor technology were seen as crucial in this process [26], issues of unlearning verification and heatmap visualization were not included.*

*Since machine unlearning and heatmap visualizations such as the metrics HC & AS are unfamiliar to organizational studies, interdisciplinary connections are needed to better understand their potential contributions. From an organizational perspective, these are some of the important questions:*

1. *How can machine unlearning and heatmap visualizations contribute to operationalizing regulations like GDPR in practice, and what implications might this have for data security and privacy?*

2. *Who should have the right to decide what gets deleted in the context of machine unlearning? What power structures or gatekeepers do this create, and how might it potentially be misused in organizational contexts?*

3. *While machine unlearning and heatmap visualizations offer tools for implementing ethical principles, such as the "right to be forgotten", how do these tools work in real-world governance contexts, where political and economic constraints are present?*

4. *What unintended consequences might arise from machine unlearning and heatmap visualizations?*

*To our knowledge, no one in organizational research has so far attempted to answer these questions. Organizational scholars, thus, currently have little to offer in terms of concrete guidelines or input for technical research or development in this context. Doing so would require closer interaction between computer science and organizational disciplines as well as analyses of concrete systems in specific application domains.*

## 6.3 A legal perspective

The following section is the full text received from the invited legal expert.

*In legal scholarship the terms "explainable" and "explainability" take on at least three distinct meanings depending on individual scholars' methodological outlook.*

*Most fundamentally, legal analysis is based on a distinction between "fact" and "law". A fact is a statement about some aspect of reality. In legal proceedings, fact is often established through a combination of methods – statements from witnesses or parties to the case, as well as various forms of supporting evidence, ranging from, e.g., background information about the issue at hand, to more technical and forensic materials, such as DNA analysis or fingerprints. As part of the latter, footage from CCTV cameras is increasingly used in legal proceedings, e.g., to identify individuals, their whereabouts or actions. Similarly, AI technologies are being used for law enforcement and evidential proceedings, e.g., to automatically track an individual's movement across multiple cameras, or to improve accuracy of facial recognitions [27]. In the example used in this article, one could imagine XAI-based analysis of harbor front monitoring being used to help "explain" fact in different types of legal proceedings. For instance, a civil lawsuit might rely on this type of analysis to document individual risk behavior or lacking safeguards in a tort case.*

*At a second level, explainability relates to the law side of the equation. The legal domain has traditionally been dominated by so-called doctrinal methods – in most parts of the world still the dominant approach to both university teaching and research [28]. Doctrinal legal research (DLR) revolves around abductive and analogical reasoning, where the correct application of legal principles and ideas are abstracted and synthesized from a variety of sources, for instance case law or legal statutes. The exact scope of these sources and their mutual relationship depends on the specific jurisdiction and field of legal specialization. Yet, DLR differs from other types of social science research insofar as it aims towards particular and non-probabilistic statements of law, as opposed to more generalizable research findings [29].*

*From this perspective, the kind of explanation that legal scholars and practitioners are concerned with is interpretive. Its aim is to distil what lawyers call "doctrine" – a set of truths accepted as authoritative. To do so, lawyers engage in a form of exegesis – a concept shared with law's theological roots, and which etymologically means exactly to explain by means of interpretation [30]. Explainability in this sense therefore remains integral to all aspects of law. Law-making is the process of explaining historically God-given or custom-based, or today more often politically decided, written rules, i.e., by clarifying these rules and their authoritative interpretation. Explainability is vice versa a necessary quality for law to be law. A fundamental requirement for rules to be accepted as law is that they must be written or displayed so that their meaning is easily understood by those persons directly affected by them. And last, but not least, explainability is a basic requirement when applying law – a legal decision must be explained and reasoned in order to be legitimate and not appear arbitrary.*

*The latter element, in particular, has gained particular importance in the context of AI-supported legal decision-making. As different forms of AI-supported or automated decision-making systems are increasingly utilizing large language models or other types of "black box" machine learning, concerns have been raised that such systems do not adhere to legal requirements for a right to an explanation. While not universally accepted and its exact scope still subject to debate, the right to (an) explanation has been codified in several jurisdictions. At the EU level, it follows from the 2016 General Data Protection Regulation (GDPR), which underscores that data subjects have the right "to obtain an explanation of the decision reached" (recital 71). More recently, the 2024 Artificial Intelligence Act (AI Act) provides a "right to explanation of individual decision-making" for high-risk systems which produce significant, adverse effects to an individual's health, safety or fundamental rights (Article 86).*

*Two opposed strategies exist for overcoming the explainability problem resulting from AI-supported legal decision-making. The first is legal and involves limiting AI-supported applications either based on pre-defined high-risk domains (as in the case of the EU AI Act) or on the basis of human-in-the-loop procedural requirements (as in the case of GDPR, which prohibits fully automated processing of personal information). The second is technical and aims to provide better explanations from existing algorithms, and algorithms that are more easily explainable – together dubbed Explainable AI (XAI). So far, however, applications of XAI in the legal domain remain limited, and no agreement exist on evaluation parameters for developing explainability acceptable to legal domain experts [31].*

*Third, and finally, recent years have seen a significant increase in more empirical legal studies (ELS), either seeking to complement or contrast doctrinal legal research. ELS have adopted a wide range of both qualitative and quantitative methods from other disciplines to the study of law and legal process. The meaning of explanation and explainability takes on different meanings depending on this methodological outlook. Qualitative empirical legal research will typically draw on interview data, observations and/or document analysis to provide explanations around how and why legal actors behave as they do, the operation of legal institutions (e.g., courts) or temporal shifts and developments in legal systems.*

*In recent years, a significant body of ELS have adopted more statistical and computational methods. Computational law aims to identify recursive patterns in legal materials by coding legal texts as data and analyzing these texts with algorithms. The majority of this work has thus far taken the form of coding written legal texts, such as judicial decisions, statutes, and international treaties, into a machine-readable form and then analyzing the texts on the basis of texts which is contained in the documents. Computational law thereby allows researchers to develop explanations that are more inductive, operate at a different scale, and offer a higher level of generalizability and controlled comparison than traditional methods.*

*As such, XAI as applied to law is uniquely suited to bridge the epistemological divide between doctrinal (DLR) and empirical (ELS) approaches marring the legal discipline. While most applications to date have been directed towards meeting legal requirements for explainability as part of automatization, XAI offers equally significant opportunities for application as part of empirical legal studies in terms of, for example, disclosing large-scale decision-making patterns that would elude individual, qualitative analysis, or identifying individual bias across multiple different dimensions. More concretely, techniques such as visual heat-mapping have recently been pioneered in the legal domain – showing promise in terms highlighting relevant words or text passages relied upon by,*

*e.g., large-language models as part of predicting legal outcomes [32]. The particular value of such techniques speaks directly to both DLR and ELS agendas – on the one hand enabling legal domain experts to manually verify AI-supported recommender systems by directing them to relevant text bits within the input material, and on the other hand disclosing, e.g., bias if relevant input highlighted turns out to correlate with formally legally irrelevant factors.*

## 6.4 An ethical perspective

The following section is the full text received from the invited ethics expert.

*Imagine that a person after having traversed Aalborg Harbor is informed that thermal cameras were in place along the harbor front, and that the cameras are being used to make ML classifications of objects on the harbor front. The person is informed that he or she was classified as a 'human being'. Having been provided this information the person responds by posing a simple question: 'Why was I classified as a human being?'*

*Assuming this to be a non-rhetorical question, the person is asking for an explanation. But what kind of explanation? Underlying the request for an explanation is an epistemic interest, i.e., an interest in getting some specific knowledge [33]. The epistemic interest picks out some set of information about a situation as relevant for a person given his or her knowledge and wider interests and values. The epistemic interests concern a multitude of aspects of a situation some of which may be classified as 'non-ethics related' and some as 'ethics related.' The person posing the question 'why was I classified as a human being' may thus be asking for many different types of explanations.*

*A non-ethics related epistemic interest could be philosophical in nature and concern the underlying model of 'humans' and 'persons.' The true meaning of the question 'why was I classified as a human being?' would therefore be something along the lines 'what are the distinctive features of a human being?' or 'what distinguishes a person from a human being?' The underlying epistemic interest could also be psychological in nature and concern the driving forces behind the tendency of people to make classifications of other people. The real question would therefore be 'what drives people to make classifications of others?' Also, the underlying interest could be technical and concern exactly what features of the thermal imaging that made the ML model make this classification. The corresponding reinterpretation of the question would consequently be 'what feature of the thermal images made the model classify me as a human being?' Additionally, the epistemic interest could also be directed at uncovering the commercial or public stakeholders facilitating the thermal camera classification of objects. The question would thus essentially be 'who is behind this implementation of thermal camera driven classification of objects on the harbor front?'*

*The epistemic interests inherent in the initial question 'why was I classified as a human being?' can also be ethics related. For present purposes, let us say that an epistemic interest is ethics related if it concerns information that is constitutive of or instrumental in protecting an individual against suffering harm or unfairness, or for protecting an individual's autonomy, privacy, dignity or some other relevant right (e.g., the right to an explanation), or for protecting other societal values such as transparency, trust, democracy, etc. To put it slightly differently, ethics related epistemic interests are interests in information and knowledge that we have ethical reasons for satisfying.*

*In the harbor case, the ethics related epistemic interests could originate in privacy concerns directly related to the classification of objects. The question 'why was I classified as a human being' would consequently be a question to the effect 'what – potentially personal and sensitive – information about me made it possible to classify me as a human being?' The question could also be related to the purpose of the classification, and the person's intention to 1) promote personal values related to that particular purpose, or 2) to protect him- or herself against potential harm and unfairness as well as autonomy and privacy violations following the use of thermal imaging for the purpose in question.*

*For the sake of argument, let us imagine that the purpose of using thermal cameras is to determine, on a daily basis, whether the level of human activity at the harbor justifies routine police patrols. Inquiring about the purpose of the classification would thus be relevant if a person believes thermal cameras to be a step in building an undesirable 'surveillance state' driven by commercial or political interests. It would also be relevant if a person is concerned that the system is inaccurate and biased*

*in ways that is likely to result in fatal absence of police patrols on the harbor front and/or less safety for particular groups of people, i.e., discrimination.*

*The upshot of these considerations is simply this. There may be a multitude of epistemic interests underlying a simple request for an explanation of AI classification, and thus a multitude of explanations (and also explanation types). Some of these interests concern information and knowledge that is constitutive of and instrumental in protecting an individual's values, rights, and wider interests.*

*As previously outlined 'heatmaps' can be used to indicate which areas of an image that have been the most important for the classification of the image. Generally, this may aid a human interpreter, but may in certain circumstances also be misleading because the heatmap does not indicate which feature in the highlighted area that was important.[34; 33]*

*Imagine now that the person requesting an explanation for his or her classification as 'human being' is presented with a 'heatmap'. Would this empower the individual to take steps to avoid suffering harm, unfairness, privacy violations as well as protect and promote his or her wider interests? A 'heatmap' may certainly be relevant in assessing the risk of loss of privacy. If the heat map shows that areas of an image with people in it are of special importance for the classification of a person, then this could indicate that personal information is being processed as part of the classification.*

*Let us alter the initial scenario slightly to incorporate the idea of privacy preserving AI. Imagine now that the person traversing the harbor is informed that thermal cameras have provided input to a classification of objects other than humans, and that the person is provided a 'heatmap' showing that no areas of a thermal image with people in it were used for the classification. Clearly this would be relevant for an assessment of the privacy violation. If no areas of a thermal image with people in it is being used for the classification of objects, then it seems hard pressed to claim that the classification violates privacy in the sense that it processes personal or sensitive information.*

*However, as our considerations in the previous sections have shown, the epistemic interests that may drive a request for an explanation are many and diverse. Narrowing the scope to ethics related epistemic interests does not change the picture. As an explanation, 'heatmaps' does little to satisfy individuals' ethics related epistemic interests, i.e., it does not empower individuals to protect themselves against using thermal imaging on the harbor front for purposes leading to discrimination and violating other deeply held values and interests. In short, 'heatmaps' have very limited value as a type of explanation that serves to satisfy ethics related epistemic interests.*

*Needless to say, our considerations in this section may have little bearing on whether to use thermal imaging for various purposes on any given harbor front. It has focused entirely on the potential of 'heatmaps' for satisfying ethics related epistemic interests.*

