# OpenReview forum: "Position: We Fool Ourselves Thinking the ‘X’ in XAI is Useful"
_NeurIPS.cc/2025/Position_Paper_Track — Submitted to NeurIPS 2025 Position Paper Track_

### Official Review · Reviewer_Ca7t · 2025-07-05

**Significance:** 2
**Presentation:** 3
**Rating:** 6
**Confidence:** 3

**Summary:**

paper argues that the dominant paradigm in explainable AI (XAI) is fundamentally flawed, because it treats explanation as a transparency or information-transfer problem rather than a contestation of epistemic authority. Drawing on a case study of machine unlearning in harbor monitoring, the authors show that stakeholder disagreements about explanations arise not from insufficient technical clarity, but from clashes over which domain has the right to validate explanations

propose a new framework, Domain Authority Negotiation (DAN), which treats explanation as an authority negotiation problem

**Strengths:**

the reframing of explanation from “information transfer” to “epistemic authority negotiation” is provocative
the harbor monitoring machine unlearning case provides a grounded, real-world demonstration of how authority clashes play out in practice
he proposed Domain Authority Negotiation (DAN) includes specific, practical elements

**Weaknesses:**

The paper is primarily theoretical and relies on a single case study, limiting the generalizability of its claims.
While DAN is described in detail, there is no discussion of how its institutional structures would actually be piloted, funded, or operationalized.
This is personal (and hence optional), but the XAI in the title is overloaded because of Musk & x.AI. Even if it is a field term, it think it best left in the body of the text not the title.

**Questions:**

How could DAN be empirically evaluated, and what metrics would show that it improves explanation outcomes compared to traditional XAI?
Could the authors specify more clearly how to handle conflicts where domain boundaries are genuinely overlapping or contested?

**Alternative Position:**

Yes, and alternative positions are well-considered and addressed by the argument

**Author Identification:**

No.

**Context:**

2

**Discussion:**

3

**Ethics:**

["NO or VERY MINOR ethics concerns only"]

**Position:**

Yes, the paper argues for or against a position related to machine learning.

**Support:**

3

**Thoroughness:**

3

---

### Official Review · Reviewer_w4mE · 2025-07-09

**Significance:** 1
**Presentation:** 2
**Rating:** 2
**Confidence:** 4

**Summary:**

This paper argues that "X" in XAI is not useful. The paper argues that useful explanations are not about information transfer and transparency. Instead, it claims that "explanation" should be viewed "as a contestation of epistemic power". The paper suggests a framework called Domain Authority Negotiation based on the literature in organizational studies. The proposed framework is based on asking three questions listed at the beginning of section 4.1.

**Strengths:**

I can't think of any strengths for the current version of this paper.

**Weaknesses:**

The position of the paper is trivial given the large body of literature already published on this topic - the literature that this position paper seems to be completely unaware of.

Literature review is unbelievably weak. Here is a paper (just one example) that criticizes AI explanations, it is highly cited, and the position paper under review is completely unaware of this literature:

Rudin, C., 2019. Stop explaining black box machine learning models for high stakes decisions and use interpretable models instead. Nature machine intelligence, 1(5), pp.206-215.

There are many papers that perform systematic studies on what explanations are needed from AI models, from the legal perspective as well as from the perspective of users, and the authorities. One example out of a large body of literature: https://arxiv.org/pdf/1711.01134

**Questions:**

Please explain your methods and procedures for literature review.

**Alternative Position:**

Yes, and alternative positions are trivial straw-man arguments

**Author Identification:**

No.

**Context:**

1

**Discussion:**

1

**Ethics:**

["NO or VERY MINOR ethics concerns only"]

**Position:**

Yes, the paper argues for or against a position related to machine learning.

**Support:**

1

**Thoroughness:**

4

---

### Official Review · Reviewer_Tzrw · 2025-08-11

**Significance:** 4
**Presentation:** 3
**Rating:** 8
**Confidence:** 4

**Summary:**

This position paper argues that the prevailing paradigm of Explainable AI (XAI) is philosophically misguided because it incorrectly frames "explanation" as a technical problem of information transparency. This work claims that the core issue is not a lack of technical tools but an unresolved struggle over "epistemic authority," where different domains like law, ethics, and computer science have competing standards for what constitutes a valid explanation.

**Strengths:**

S1: A significant strength of the paper is that it moves beyond critique to offer a constructive, actionable alternative to current XAI practices. The proposed Domain Authority Negotiation (DAN) framework reframes the problem from one of technical transparency to one of socio-technical governance, focusing on mapping authority domains, defining their boundaries, and establishing mechanisms for negotiation. This provides a new vocabulary and a structured approach for addressing the deep-seated interdisciplinary conflicts the paper identifies. Though I have some comments regarding DAN, I will reserve them for O1.

S2: The paper’s central argument is substantiated through a concrete, qualitative study involving experts from organizational studies, law, and ethics who evaluate a real-world XAI application. This methodological approach provides an empirical grounding for the theoretical claim that conceptual misalignment stems from competing authority claims. By presenting and analyzing the distinct responses from each expert, the paper makes the abstract concept of "epistemic power dynamics" tangible and more convincing.

**Weaknesses:**

O1: While the DAN framework is conceptually robust, the paper provides limited detail on how its institutional structures, such as "Authority Councils" and "Negotiation Protocols," would function in practice. The potential for these mechanisms to introduce significant organizational overhead, slow down development, or become bogged down in bureaucratic disputes is not fully explored. The paper acknowledges that DAN could entrench existing power imbalances if not carefully implemented, but the safeguards against this outcome remain theoretical.

O2: The core empirical evidence is derived from the detailed responses of one expert from each of the three chosen disciplines (organizational studies, law, and ethics). The paper itself notes that a different legal expert, such as one specializing in privacy law, might have provided a very different response. This limited sample makes it difficult to generalize the findings as representative of entire disciplines, as the views presented may reflect the specific perspectives of the individuals rather than a broader disciplinary consensus.

**Questions:**

N/A

**Alternative Position:**

Yes, and alternative positions are well-considered and addressed by the argument

**Author Identification:**

No.

**Context:**

3

**Discussion:**

4

**Ethics:**

["NO or VERY MINOR ethics concerns only"]

**Position:**

Yes, the paper argues for or against a position related to machine learning.

**Support:**

4

**Thoroughness:**

4

---

### Note · Authors · 2025-09-02

**1-10 Additional Comments:**

The variance in review quality was striking: while two reviewers engaged substantively with our position, one appeared to evaluate it as a technical critique rather than the philosophical and institutional reconceptualization we intended. This underscores the value of clearer reviewing guidelines for position papers. That said, we view such variance as a success in itself — a position paper’s role is to spark debate and engagement across diverse perspectives, including those shaped by different disciplinary power dynamics.

**1-11 Submit Again:**

Probably yes

**1-1 Submission Process:**

4

**1-2 Next Year:**

Great to have a position paper track. We would like to see more structured dialogue between position paper authors and reviewers during the review period, though. Position papers benefit from intellectual engagement rather than just evaluation.

**1-3 Future Development:**

Consider providing reviewers with explicit guidance on how to evaluate position papers differently from technical contributions. Criteria should emphasize novel conceptual contributions and argumentative clarity, rather than exhaustive literature coverage.

**1-4 Interest:**

["Panel discussions with other position paper authors", "Structured debates on controversial topics", "Workshops for developing position papers", "Mentorship programs for early-career researchers"]

**1-5 Thoughtful:**

7

**1-6 Supportive:**

6

**1-7 Technical Aspects Versus Position:**

8

**1-8 Gate Keeping:**

6

**1-9 Camera Ready Changes:**

We will add citations to clarify our awareness of related work while maintaining the philosophical nature of the paper and that our contribution addresses an orthogonal problem. We'll also acknowledge implementation limitations as appropriate for a position paper introducing a new conceptual framework.

**3-1 Review Response1:**

Tzrw

**3-2 Reaction To Review1:**

We thank Reviewer #1 for the very positive review and for recognizing the intention of our position paper. Below are a few remarks addressing the objections:

O1 (Implementation details): We appreciate this comment. As a position paper, our primary goal was to identify the problem and outline the solution space. A detailed implementation would necessarily involve empirical pilots and iterative development with real institutions, which lies beyond the scope of a conceptual contribution of this kind.

O2 (Sample size): Thank you for this thoughtful observation. Our methodology, however, was not designed for statistical generalization, but rather as a proof of concept: to demonstrate that different domains interpret the same XAI outputs through incompatible authority lenses. Even with only three experts, we observed substantial conceptual misalignment and competing authority claims. Moreover, the intra-disciplinary variance (e.g., between privacy and tort law, as the reviewer notes) reinforces our claim that authority conflicts emerge not only between disciplines but also within them. Thus, the conceptual contribution of the paper remains valid and independent of these empirical limitations.

**3-3 Review Response2:**

w4mE

**3-4 Reaction To Review2:**

We interpret the strongly negative stance of this review (e.g., “trivial straw man arguments,” “unbelievably weak”) as reflecting a disciplinary mismatch rather than a substantive evaluation of our paper’s aims. Our work addresses a deep philosophical issue: conceptual misalignment in XAI. This problem extends far beyond communication challenges and manifests as struggles over epistemic authority.

The claim that we are “completely unaware” of key literature (e.g., Rudin) is a misunderstanding. While the field has often framed stakeholder diversity as a technical challenge to be solved with algorithms, user-centered methods, or communication frameworks (see Langer+ 2021 [stakeholder]; Longo+ 2024 [XAI 2.0]), our argument is orthogonal: such approaches treat explanation as mere information transfer, overlooking the underlying conflicts of epistemic authority. Institutional mechanisms—not technical fixes—are required.

The references identified as “overlooked” do not directly address this point. Their use as decisive counterarguments in fact illustrates our claim: different traditions valorize different authorities, underscoring the disciplinary tensions at stake.

We agree that XAI has been widely criticized, with repeated calls for more interdisciplinary work on interpretability and explainability. Our contribution shows, however, that such discussions remain fundamentally futile without first addressing power dynamics and philosophical misalignments. We therefore propose DAN as a framework for defining and delivering what truly explainable AI requires.

Finally, we respectfully disagree with the characterization of our argument as “trivial.” Recasting misalignment as authority conflict reframes the entire XAI discourse. If this point were trivial, it would already be pervasive in the literature; it is not. In this sense, the review itself demonstrates the need for our position paper: differing standards across disciplines are precisely the problem we diagnose.

**3-5 Review Response3:**

Ca7t

**3-6 Reaction To Review3:**

We thank Reviewer #3 for the positive review and for recognizing the intention of our position paper. We also appreciate the reviewer’s acknowledgment of the relevance of the case study as well as the paper’s potential to provoke discussion.

Below are a few remarks addressing the objections:

O1: (Theoretical and only a single case study): We agree that our argumentation is primarily philosophical, given the nature of the issue addressed. Expanding the study to include multiple cases would certainly be valuable, but such an empirical project would go beyond the scope of a position paper. Our aim here is to identify and frame the underlying conceptual problem. Demonstrating the issue clearly in a single case provides a strong foundation for establishing discussion about its broader scope and significance.

O2: (No discussion of operationalisation): As with review #1, We appreciate this comment. Again, our primary goal was to identify the problem and outline the solution space. A detailed implementation would necessarily involve empirical pilots and iterative development with real institutions, which lies beyond the scope of a conceptual contribution of this kind.

---

### Meta-Review · Area_Chair_6GYC · 2025-09-14

**Rating:** 4
**Confidence:** 5

**Strengths:**

The authors offer a compelling perspective on shifting the framing of XAI from a purely technical, transparency-oriented approach to a broader socio-technical one. They support this position with insights drawn from a qualitative study.

**Weaknesses:**

The main limitation lies in the strength of the evidence supporting the authors’ position. While the use of a qualitative method is appreciated for its potential to provide empirical grounding for the theoretical claims, the study itself is based on non–peer-reviewed work, and reviewers expressed concerns about the small number of experts involved. They also noted insufficient engagement with the existing literature supporting similar perspectives. Given the breadth of research on XAI, unlike in more emerging areas, the concern about conducting a thorough literature review to substantiate the identified gaps appears well justified. Additional concerns were raised regarding the lack of detail on the challenges related to DAN’s operationalization.

**Questions:**

Reviewers raised important questions regarding how DAN could be empirically evaluated, as well as the challenges involved in its operationalization.

**Ethics:**

No ethical violations or concerns were raised by the reviewers.

**Thoroughness:**

5

---

### Decision · Program_Chairs · 2025-09-26

Reject